Home range and use of diurnal shelters by the Etendeka round-eared sengi, a newly discovered Namibian endemic desert mammal

Rathbun Galen B. grathbun@calacademy.org
Dumbacher John P.
Institute of Biodiversity Science and Sustainability, California Academy of Sciences , San Francicso, CA , United States
Kramer Donald
Electronic publication date: 2015 Oct 1
Publication date: 2015
Volume: 3
Electronic Location ID: e1302
Received 2015 Jul 13; Accepted 2015 Sep 16
Copyright: © 2015 Rathbun and Dumbacher
Copyright year: 2015
Copyright holder: Rathbun and Dumbacher
License: This is an open access article distributed under the terms of the Creative Commons Attribution License, which permits unrestricted use, distribution, reproduction and adaptation in any medium and for any purpose provided that it is properly attributed. For attribution, the original author(s), title, publication source (PeerJ) and either DOI or URL of the article must be cited.
License URL: https://creativecommons.org/licenses/by/4.0/

Keywords: Elephant-shrew, Home range, Shelter, Namib Desert, Namibia, Sengi

Funding: The authors received no funding for this work.

==============================
To understand habitat use by the newly described Etendeka round-eared sengi (Macroscelides micus) in northwestern Namibia, we radio-tracked five individuals for nearly a month. Home ranges (100% convex polygons) in the rocky desert habitat were remarkably large (mean 14.9 ha) when compared to sengi species in more mesic habitats (<1.5 ha). The activity pattern of M. micus was strictly nocturnal, which contrasts to the normal diurnal or crepuscular activity of other sengis. The day shelters of M. micus were under single rocks and they likely were occupied by single sengis. One tagged sengi used 22 different day shelters during the study. On average, only 7% of the day shelters were used more than once by the five tagged sengis. The shelters were also unusual for a small mammal in that they were unmodified in terms of excavation or nesting material. Shelter entrances were significantly oriented to face south by south west (average 193°), away from the angle of the prevailing midday sun. This suggests that solar radiation is probably an important aspect of M. micus thermal ecology, similar to other sengis. Compared to published data on other sengis, M. micus generally conforms to the unique sengi adaptive syndrome, but with modifications related to its hyper-arid habitat.

Introduction

The sengis or elephant-shrews (Order Macroscelidea) are a well-defined monophyletic clade of mammals that are endemic to Africa, not closely related to other clades in the supercohort Afrotheria (Seiffert, 2007). There are only 19 extant species, which are divided into the subfamilies Rhynchocyoninae and Macroscelidinae (Corbet & Hanks, 1968). The four species of Rhynchocyon in the first subfamily are forest dwellers in central and eastern Africa and weigh between 300 and 750 g (Rovero et al., 2008). The genera Petrodromus, Elephantulus, and Macroscelides are in the second subfamily. Petrodromus is monospecific, weighs about 200 g, and occupies thickets, dense woodlands, and forests of central and eastern Africa (Jennings & Rathbun, 2001). The 12 species of Elephantulus (Smit et al., 2008) weigh from 45 to 60 g, occupy habitats that include grasslands, bushlands, and open woodlands throughout much of Africa, with the exception of the Sahara Desert and western Africa (Rathbun, 2015). The three species of Macroscelides occur in the deserts of southwestern Africa, and weigh only 25–45 g (Dumbacher et al., 2014).

From the earliest studies of sengis (Sauer, 1973; Rathbun, 1979), it was recognized that their combined life history traits formed a unique adaptive syndrome, not seen in any other mammals in other biogeographic regions of the world. The syndrome blends life history strategies usually associated with ant-eaters and some antelopes, including a diet of invertebrates with an associated long nose and tongue and small mouth, highly cursorial locomotion, small precocial litters, absentee maternal care, lack of nest-use (Macroscelidinae only), and social monogamy. These traits do not vary greatly among the species so far studied, despite the considerable variation in their size and habitats (Rathbun, 1979; Rathbun, 2009).

When it was found that some sengis were socially monogamous (Rathbun, 1979), which is unusual in mammals (Komer & Brotherton, 1997), additional studies were completed to better understand the evolution of this social organization (FitzGibbon, 1995; Ribble & Perrin, 2005; Rathbun & Rathbun, 2006; Schubert et al., 2009; Oxenham & Perrin, 2009). One of the main focuses of these studies has been home range characteristics, but other aspects of their life history have been documented incidentally, such as the unusual sheltering habits among the Macroscelidinae.

Although Rathbun (2009) reviewed sengi taxonomy and life history traits, recent taxonomic revisions have resulted in new taxa being recognized (Rovero et al., 2008; Smit et al., 2008; Dumbacher et al., 2012). The Etendeka round-eared sengi (Macroscelides micus Dumbacher & Rathbun, 2014) was discovered in 2006 and is the newest species to be described (Dumbacher et al., 2014). It is the smallest sengi and only occurs in a small remote hyper-arid area in northwestern Namibia, sandwiched between the coastal Namib Desert and the inland escarpment (Swart & Marais, 2009; Rathbun, Osborne & Coals, 2015).

The objective of our research on M. micus was to gather the first basic information on habitat and shelter use to determine the similarity of these and other life history traits to those of previously studied sengis, especially the Macroscelidinae.

Materials and Methods

Our study site (latitude −21.32338, longitude 14.32738) was in northwestern Namibia, within the eastern edge of the Namib Desert, and the lower eastern slope of the Goboboseb Mountains, which are part of the Etendeka geological formation that was created by lava flood events about 132 million years ago (Swart & Marais, 2009; Fig. 1). The study site was about 580 m above sea level, on the lower slopes of a 900 m high mountain. The slopes (average = 13.4°, range = 3–29°, N = 48) were composed of rust-colored compact gravel with an estimated 40–95% of the surface covered with fist to building-block sized rocks (Fig. 1). The closest town was Uis (population about 4,000), about 60 km to the east. The study site was about 55 km inland from the cold Benguela ocean current, which resulted in wet coastal fogs at our site on about a quarter of the nights. The fog left moisture on rock surfaces, but both completely dissipated by midmorning. Based on our interpolation of weather data from Henties Bay and Uis, we estimate the average yearly rainfall at the study site is 10 mm. During our fieldwork, the average overnight low temperature at our study site was 9.6 °C (range = 3.9–18.7 °C), and the average maximum (afternoon) temperature was 27.8 °C (22.0–30.0 °C). On many afternoons, winds blew up to 13.5 m/s (48 km/h). Full moon occurred on 7 October, and sunrise and sunset was at about 0630 and 1905 h respectively.

Figure 1 Study site in eastern Goboboseb Mountains, northwestern Namibia.

View from the northern end of #4947M home range looking south across home range area of #4020F (see Fig. 3). White flagging on top of rock in foreground is a day shelter of #4947M. Boscia bush in far middle of image was used as a shelter at night (see ‘Results’). The alluvial plains between the Boscia bush and the sand dunes in far distance, beyond rust-colored rocky Macroscelides micus habitat in foreground, were rarely used by M. micus, but are likely habitat of M. flavicaudatus. Wooden handle of radio-tracking antenna on right margin of image is 30 cm long. Photo 23 October 2014 by GBR.

Our study spanned from 30 September through 26 October 2014, and we trapped (HB Sherman Traps, Tallahassee, Florida; model LFA, 7.6 × 8.9 × 22.9 cm) and tagged sengis on 13 days during the first two weeks. We set about 200 traps per night at 10–20 m intervals on transects within about 50 ha of likely M. micus habitat, and traps were moved to new transects every 1–4 days. We baited traps with a dry mixture of rolled oats, peanut butter, and Marmite (a yeast paste or spread), opened the traps at dusk, and checked and closed them at dawn. Trapped sengis (we only captured M. micus) were immediately tagged and released at the capture site. At the end of our study, the sengis were recaptured at their day shelters by hand or flushed into mist nets (DTX 36 mm stretch mesh), all radios and tags were removed, and the sengis were released.

We attached a reflective ear-tag inside the distal margin of a pinna—right ears of males, and left ears of females. The tags were constructed of two 5-mm-diameter disks of reflective silver-colored plastic (Reflexite FD 1430 marine adhesive tape), which only reflected when a light source was aligned closely with the spotter’s eyes, thus eliminating the likelihood of increased predation on the ear-tagged sengis on moon-lit nights. The disks were attached to an ear with a nylon stud (monofilament fishing line) through holes previously melted in the centers of the two disks and a hole pierced through the pinna (Fig. 2A; see Rathbun, 1979; Rathbun & Rathbun, 2006 for further details). Because the sengis were nocturnal (see ‘Results’), we used bright (275 Lumen) narrow-beamed light-emitting diode (LED) headlamps (Princeton Tec model Apex, and Fenix model HP15) to spot the ear-tagged sengis, often with the aid of binoculars. The ear tags were visible up to 100 m away, but fog, dust, and rocks often reduced visibility to much lower distances. Vegetation was sparse or lacking and did not hinder visibility (Fig. 1).

Figure 2 Ear-tagged and radio-collared M. micus at study site in Goboboseb Mountains, Namibia.

(A) Sengi #4856F under Commiphora bush on 22 Oct 2014 at 2342 h. Visible are the reflective tag on left ear and transmitter antenna extending from top of neck over back. Radio collar is completely hidden by fur. (B) Sengi #4947M at the opening of a typical rock shelter on 25 Oct 2014 at 2351 h. Photos by GBR.

We also attached radio-collars (Holohil Systems; Carp, Ontario, Canada; transmitter model BD-2C, frequencies in 164 MHz band, weight about 1.5 g) to seven of the eight captured sengis. The transmitter whip antennae were incorporated into Tygon tubing collars, leaving about 8 cm extending from the top of the collars, and the transmitters hung from the bottom of the collars.

We located the radio-tagged sengis by homing (Kenward, 2001) using receivers (model R-1000; Communications Specialist, Orange, California, USA; model R-TRX-1000S; Wildlife Materials International, Murphysboro, Illinois, USA) attached to two-element Yagi directional receiving antennae (Telonics, Mesa, Arizona, USA). Upon approaching a sengi, the ear-tag was easily spotted, when we made a mental note of a prominent landscape feature at the sengi’s location, and took GPS coordinates there. If the sengi was sheltering under a rock, we took the coordinates of the shelter. At night, one of us radio-tracked from about 2100 h to 0100 h, and the other from about 0200 h to 0600 h. Combining our effort, each sengi was located between two and six times per night, in arbitrary order. During the day, we located sheltering animals in the morning or midday, and again at dusk when we monitored the departure of selected sengis from their day shelters. We determined universal transverse Mercator (UTM) coordinates with the GPS functions on a Motorola MotoG (2013 model) mobile phone and a Samsung Galaxy Player 4. Both receivers used the Android operating systems with the LOCUS MAPS navigation application (version 3.4.0) for entering, storing, plotting, and exporting location coordinates and associated data. In the field, locations were based on 1 s intervals averaged during 15–60 s. We tested the accuracy of the receivers at the field site, and they were within a diameter of 5 m (MotoG) and 10 m (Galaxy).

To determine home range areas, we used RANGES 9 software (Anatrack Ltd., Wareham, Dorset, UK). We ran several different analyses (Kenward, 2001) in order to compare home range size estimates with published values. We included the object restricted-edge polygon (OREP) analysis (Anatrack, 2015) because this and the concave polygon analysis may be useful for future comparisons. For the analyses, we used a censored data set that included capture localities (except for the OREP analysis), all radio and sighting records, all day and night shelter locations, and the final capture (or death) location for each individual. We eliminated records that were obviously incorrect due to observer error. Because we have not analyzed the data for differential use of home range areas, and the sengis were remarkably active and swift during the night, we did not censor the data set for location and time auto-correlations. For all home range analyses, the units of measure were meters with the resolution set at 1 m, and we used the ‘curve and polygon’ option in RANGES 9. To keep our home range estimates comparable to published estimates, we only used the ‘buffer tracking resolution’ option for the concave polygon and OREP analyses. For the convex polygon (= minimum convex polygon or MCP) analysis we used 95% and 100% ‘cores’ based on ‘arithmetic mean centers.’ For the concave polygon analysis we used the ‘selected edge restriction’ option with a value of 0.4. For the OREP analysis we used the ‘>5% distribution distance’ and ‘KED and Strip’ options. We used all the default settings for the 95% core kernel analysis, which were fixed kernel, location density contours, fixed smoothing multiplier, and 40 matrix cells set to rescale to fit matrix.

While radio-tacking sengis after dawn, we located, flagged and recorded GPS coordinates for the day shelter used by each sengi, and then rechecked shelters arbitrarily during the remainder of the day for continued occupancy. The last and most focused check started at about sunset, when one of us sat inconspicuously among rocks or boulders about 5–10 m from an occupied shelter, and watched for sengi movement and listened for variations in radio signal pitch, strength, and direction, which indicated an active sengi. Once the animal was active, we briefly searched the area around the shelter with binoculars and headlamp for an ear-tag reflection, thus further confirming that the sengi was active and had departed its shelter for the night.

Near the end of the field study, we sampled sengi day shelters and took a set of standardized metrics that included the orientation of the rock shelter entrance, gross habitat characteristics (aspect, slope, ground cover), midday ambient air temperature, temperature inside the shelter, and temperature on the top surface (facing the sun) of the shelter rock. We also measured the dimensions of the rock forming the shelter (approximate length, width, and vertical thickness). We then carefully removed and then replaced the shelter rock to record the substrate inside the shelter (gravel, sand, dust), and looked for evidence of occupation (excavation, presence of bedding, or feces).

We recorded the various temperatures because the dark rust-colored rocks heat up from direct solar radiation based largely upon the area of rock that is exposed to the sun (length and width of rock). The thermal inertia of the rock will be approximately linearly related to its thickness (or mass of the rock divided by the surface area exposed to the sun). We therefore regressed measures of shelter temperature against the shelter rock thickness to test whether thicker rocks provide more stable temperature environments and protection from midday heat.

Our study was approved by the Namibia Ministry of Environment and Tourism (permit number 1927/2014), and reviewed by the California Academy of Sciences Institutional Animal Care and Use Committee (approval number 2014-1).

Results

Capture and radio-tracking

We accumulated 2,742 trap-nights, capturing 3 rodents (one each of Gerbillurus, Petromyscus, Petromus; 0.11% trap success) and 7 M. micus individuals (0.26%). To try to capture all the sengi individuals at the study site, we often set trap transects across areas where we had already captured sengis, in addition to adjacent areas. Remarkably, we only once recaptured one of our tagged sengis (#4612F). We captured an eighth sengi by hand at night (#4585M), but only collared seven (Table 1); a single young female was only ear-tagged. Both #4427F and #4585M disappeared soon after collaring, and provided no data. For any particular analysis, a subset of only relevant data were used, thus sample sizes did not always conform to the overall totals shown in Table 1.

Table 1 Data associated with Etendeka round-eared sengis captured at the study site in the Goboboseb Mountains, Namiba.

ID	Sex	Age	Wt (g)	Initial capt date	Fate at end of study	Fate date	Total days radio-tracked	
4020	Female	Adult	31.5	30 Sept	Released	26 Oct	27*	
4254	Male	Adult	–	8 Oct	Released	26 Oct	17*	
4427	Female	Young	16.0	8 Oct	Disappeared	9 Oct	1	
Ear tag	Female	Young	16.0	8 Oct	Disappeared	13 Oct	–	
4585	Male	Adult	26.5	15 Oct	Disappeared	16 Oct	1	
4612	Female	Adult	–	10 Oct	Predation	15 Oct	5*	
4856	Female	Adult	34.0	3 Oct	Released	26 Oct	24*	
4947	Male	Adult	31.0	3 Oct	Released	26 Oct	24*	
Notes.

* Only those sengis with an * in last column were used for home range analyses.

Home range

The average home range sizes of the five radio-collared sengis, as determined by the different methods of analyses, were highly variable (Table 2), spanning from 7.2 to 22.8 ha. The average maximum length of the home ranges, calculated using the 100% convex polygon method, was 705 m. However, this was greatly influenced by #4254M that had a remarkably large oblong-shaped home range (Figs. 3 and 4). The average distance between the arithmetic mean centers of overlapping 100% convex home ranges was 425 m, range 256–608 m, which is a useful comparative measure of sengi dispersion (see interpretation of M. flavicaudatus home range estimates in Discussion). Because of our small sample size and uncertainty that all resident sengis in the study area were captured, we have not presented detailed overlap analyses.

Figure 3 Object restricted-edge polygon (OREP) home range polygons for five radio-collared Macroscelides micus at study site in Goboboseb Mountains, Namibia.

See Table 2 for home range areas. Note the disjointed home range of #4856F, with two points within the home range of #4947M. Home range polygons (colored for clarity) are concentrated on lower rocky slopes of rust-colored Etendeka volcanic substrate, with the exception of #4254M and #4020F (see ‘Results’ and Fig. 1). Background satellite image captured on 17 Aug 2004, © 2015 Google Earth, DigitalGlobe.

Figure 4 Home range polygons for five radio-collared Macroscelides micus at study site in the Goboboseb Mountains, Namibia.

Colors and identifications same as Fig. 3, see Table 2 for areas. (A) minimum convex polygons for home ranges (solid lines based on 100% of points) and day shelters (dashed lines 100% of shelters). Initial capture locations are shown with a star that match individual home range line colors. Capture locations of young #4427F and ear-tagged female are shown with a blue X, and adult #4585M in a blue triangle (see Table 1). (B) Kernel 95% contour home range areas (see Table 1), including stars at initial capture locations.

Table 2 Home range areas (ha) of five radio-collared sengis (see Table 1) at the Goboboseb Mountains study site in Namibia.

Sengi ID	Obs no	CP 100%	CP 95%	Kernel	Concave 0.4	OREP	Max distance	% day shelter area	
4020F	102	8.48	5.35	5.64	5.0	6.46	549	25.7	
4254M	56	36.21	34.05	82.81	16.0	13.44	1,619	90.6	
4612F	18	5.5	4.13	8.58	2.4	2.42	371	24.4	
4856F	89	17.22	9.44	10.16	10.4	6.49	619	13.3	
4947M	92	7.23	5.23	6.6	5.77	7.28	367	30.0	
Average	–	14.92	11.64	22.76	7.91	7.21	705	36.8	
Notes.

Column headingsObs No number of locations used in home range analyses

CP convex polygon with 100% and 95% of locations

Kernel kernel with 95% locations

Concave concave polygon with 0.4 edge restricted option

OREP objective restricted-edge polygon (see Methods)

Max distance maximum distance across CP 100% home range in meters

% day shelter area proportion of shelter area to home range area based on 100% convex polygon estimates

The home range size (Table 2) and shape (Figs. 3 and 4) of #4254M was odd compared to the other four sengis. We located this male mostly at each end of his oblong-shaped home range, which spanned over 1.5 km (Table 2). He moved from end-to-end of his home range 11 times, making the journey so quickly that we were only able to roughly track his path once, when he traveled the length of his home range (about 1.5 km and over 80 m in elevation) within 60 min, presumably in a relatively straight course with few pauses. The area between the ends of his home range was atypical habitat for M. micus, being a slightly sloping alluvial fan composed of softer and lighter-colored gravels and fewer rocks than on the surrounding higher slopes (Fig. 3). The only other home range that was not completely located on rust-colored Etendeka volcanic substrates was that of #4020F, with about 0.73 ha at the southern edge falling on the lowest alluvial flats in the study area, which were composed of finer and lighter colored gravels with few rocks on the surface (Figs. 1 and 3). The home range areas of all the sengis tended to fall below the steeper areas of the Etendeka formation that had huge boulders and rock faces (Fig. 3).

We closely followed #4020F on her home range twice during the night of 1 October 2014 by keeping sight of her reflective ear-tag. Starting at 2152 h, she covered about 219 m in 10 min (1.3 km/h) and her route (based on the GPS-determined track of the observer) was a large circle that did not quite meet the starting point. The second track started at 2217 h, and covered 89 m in 3 min (1.6 km/h) in roughly a straight line. The sengi easily kept ahead of us as it bounded from rock to rock, obviously following a familiar route. During our study, we found no worn sengi paths across the substrate, but nevertheless they appeared to follow familiar routes, as demonstrated when we spotted a lone unmarked sengi (became #4585M) within the home range of #4856F. The sengi appeared unfamiliar with the area because he continually stumbled over and bumped into rocks as he clumsily fled, which allowed us to chase and hand-capture him while keeping him in the beam of our headlamp (no physical or visual impairments were noted when collared and released). It was impossible to similarly capture our tagged sengis because they were too agile and swift.

After we radio-collared #4585M, we only located him once the next day, even though we searched widely (several km) in areas adjacent to our study area on several days. Because our transmitters had a line-of-sight range of about 1 km, it seems unlikely that we lost the signal. It is possible that the transmitter failed, but we never spotted any male ear-tagged sengis without an associated radio signal. Sengi #4585M possibly became prey of a Cape fox (Vulpes chama A Smith, 1833) that we saw in our study site on several nights. This was probably also the fate of #4612F, given that we found her shed and functioning transmitter with tooth damage (Table 1).

The areas (100% convex polygons) encompassing all day shelters for each of the five radio-tagged sengis averaged 36.8% of their respective home range (Table 2). The distribution of day shelters within a home range showed no obvious pattern, other than the sengis used locations with suitable rock shelters and tended to be well inside the home range boundaries (Fig. 4).

Shelter characteristics

We examined a sample of day shelters used by the five collared sengis (#4020F n = 13, #4254M n = 9, #4612F n = 5, #4856F n = 11, #4947M n = 11). The ground surrounding the shelters was always rock and boulder strewn, averaging 52% coverage (range 40–95%). Aspect and slope varied by animal, but showed no overall trend that differed from the surrounding habitat in each home range. Shelters were typically under a single rock with a horizontal crevice (Fig. 2B) with an average height of 6.6 cm (range 3–12 cm). No shelters showed any obvious signs of alteration, such as excavation, digging, or collected bedding. Three of 49 shelters had some windblown grasses or plant matter, but it was never noticeably arranged or manipulated, and seemed typical of the surrounding boulder fields. Interior substrates of dust, sand, or gravel more or less matched the surrounding substrate. Only one shelter of 49 contained feces (3 pellets), and none of the shelters had partially eaten food or scraps. The entrances to shelters showed significant directionality (Raleigh’s Z test, n = 41, z = 3.66, p < 0.05) with an average compass direction of 193° south by southwest, despite the fact that slope aspect varied among individuals and showed no overall directionality (Raleigh’s Z test, n = 49, z = 0.35, p > 0.2).

We regressed shelter temperature against the shelter rock thickness and recovered a significant negative relationship (n = 47, R2 = 0.396, p < 0.01, Fig. 5A), thus confirming that thicker rocks may provide more stable temperature environments and protection from wide temperature fluctuations. Because we measured shelters on different days, we additionally sought to control for differences in midday temperature by subtracting shelter temperature from local ambient air temperature. We again found a significant positive relationship, suggesting that thicker rocks were cooler relative to air temperature (general linear regression, n = 47, R2 = 0.4117, p < 0.01, Fig. 5B). Despite confirming the potential benefit of thicker shelter rocks to protect from extreme temperatures or wide fluctuations, we cannot confirm whether sengis are actually choosing shelters to take advantage of these benefits. In fact, most shelters were under rocks with smaller thicknesses (Fig. 5), but it is not clear whether this is due to an active choice on the part of sengis, or whether they are constrained by availability.

Figure 5 Regressions investigating the thermal inertia of Macroscelides micus shelter rocks.

Graphs illustrate the negative relationship between shelter temperature and the thickness of the shelter rock (A). Because we measured shelters on different days, with different ambient air temperatures, we also plotted (B) the difference between air temperature and shelter temperature and regressed this against rock thickness.

Shelter use

The collared sengis were strictly nocturnal. Once sheltered at night, usually near dawn, they normally remained in the same shelters throughout the day, and were very reluctant to leave. For example, we checked 33 occupied shelters twice during the day between 13 and 5.5 h prior to sunset, and in only two cases did a sengi change shelters. In one case (#4947M) the distance between shelters was about 3 m, and in the second case (#4612F) it was about 30 m. On three days we checked #4020F four different times during daylight, and on one day three times, and #4856F at four different times on one day. Neither of these sengis shifted shelters during the day. When we recaptured the four remaining radio-tagged sengis at the end of the study on 26 October 2014, between 1000 h and 1145 h, we had to dislodge or remove the shelter boulders to get the animals to flee into the capture nets, which further demonstrated their reluctance to leave their day shelters.

We never observed or radio-tracked any diurnal sengi movements, and they all were active on every night with one exception. During the night of 8 October, #4020F did not leave her day shelter, and when we checked her after dawn she was torpid in her shelter. We thought she might have entangled a forefoot in her collar, but upon capture we found no problems. She quickly came out of torpor and after her release she resumed her typical nocturnal activity pattern.

We determined whether collared sengis used different shelters during late night (usually just before dawn) compared to the following day. In 26 of the 31 cases, switching did not occur, indicating that the sengis often sheltered for the day well before first light. Related to this pattern, we extracted location data for four sengis (those with the most robust overall data sets: #4020F, #4254M, #4856F, #4947M) and determined whether we found them in a night shelter or not during two periods: between 2100 and 0100 h (early night), and between 0200 and 0600 h (late night). In the early period, there were 130 pooled observations, with 14 in night shelters (10.8%). During the late period, we had 117 observations with 37 (31.6%) in night shelters. These data support our subjective assessment that the animals were more active early in the night compared to late at night. This pattern made it nearly impossible for us to determine when animals retreated to shelters for the day, compared to when they left their day shelters for a night of activity. We monitored 40 day shelters starting at about sunset and the average departure time was 1938 h, with a range of 1913–1959 h. In two additional cases, a sengi (#4254M) had not left the day shelter by 2010 and 2015 h, when we terminated observations.

Even though the sengis rarely switched shelters within a day, they readily switched shelters from day to day, rarely using a site more than once (Table 3). Pooling individuals, we monitored 85 day shelters and 93% were used once, 5% twice, and 1% each for three and four times. The average interval between using the same shelter was 3.2 days, with a range of 1–9 days. We found no evidence that different individuals used the same shelter, nor that more than one sengi occupied a shelter at the same time, although it is remotely possible that untagged sengis paired with our collared animals.

Table 3 Day to day shelter use by five radio-tagged sengis at the Goboboseb Mountains study site in Namibia.

Columns labeled “used…” are the number of times different day rock shelters were used during the study period by each individual (see text). The “Day intervals” column indicates the number of days between sequential use of the different shelters (separated by a slash). For example, 4020F used three different shelters twice each, and the days between the use of each of these shelters was 1, 6, and 1 days. This same sengi used one shelter four times, with the intervals between each use (separated by commas) being 3, 3, and 1 days. The total number of unique shelters used for each individual is in last column.

Sengi ID	Used × 1	Used × 2	Used × 3	Used × 4	Day intervals	Total	
4020F	18	3	0	1	1/6/1/3,3,1	22	
4254M	16	0	0	0	–	16	
4612F	5	0	0	0	–	5	
4856F	19	0	1	0	4,1	20	
4947M	21	1	0	0	9	22	
Total	79	4	1	1	–	–	

Twice we located sengis under low bushes at night—a 1 m high Commiphora bush and a 2 m high Boscia bush (Fig. 1). Bushes in this size range only numbered 2 or 3 individuals in each home range. While under the canopy of these bushes, the sengis were “nervous” and easily disturbed by our movement, running to the opposite side of the bush from the observer on several occasions, but they did not flush into the open nor did they foot-drum. While we were about 5 m from the animals, we observed them for about 15 min while they groomed and rested (Fig. 2A) on the surface of the gravel substrate. They were always alert with their eyes open and ready to flee. These observations were terminated after they bounded off into the night.

Discussion

We were unsure what to expect in a first study of the behavioral ecology of a newly discovered species found only in a hyper-arid desert, especially in a group of mammals already known for several unusual traits (see ‘Introduction’ and Rathbun, 2009). Soon after starting our field work, we realized that densities were disappointingly low, and in conjunction with the difficult logistics associated with large home ranges, we knew our data would be limited. Faced with interpreting our findings, we realized that few insights could be gained in terms of general home range information and theory. However, some of our findings provided ample opportunity to compare some features among different sengi species, and thereby gain some interesting insights into the behavioral ecology of this group. Below, we discuss our findings in the context of sengi life history traits that lead to further insights into the sengi adaptive syndrome.

Home ranges

The estimated home range sizes of the five sengis we collared are dependent on the method of analysis. Both convex polygon and kernel methods incorporate areas that were rarely if ever used, but we have included both metrics to allow comparison with published data. Our limited data suggest that a more conservative representation of the home ranges of our tagged sengis is obtained with the relatively new OREP method, especially for #4254M (Table 2, Figs. 3 and 4). Unfortunately, no previous sengi studies have used this method, as is the case with the concave polygon technique. We nevertheless have included both with the hope that future studies will also find that they are a useful alternative to convex polygon estimates because they may provide better insights into space use by sengis.

The three species of Macroscelides occupy very arid habitats (Dumbacher et al., 2012; Dumbacher et al., 2014) compared to other sengis, thus ecological insights may be gained from these and other sengi species. Schubert et al. (2009) provides quantitative home range data for the Karoo round-eared sengi (Macroscelides proboscideus Shaw 1800) near Springbok, South Africa, based on radio-tracking methods. Using direct observations, Franz Sauer (1973) with his wife Elinore report home ranges of about 1 sq km for the Namib round-eared sengi (Macroscelides flavicaudatus Lundholm 1955) in the Namib Desert southeast of Walvis Bay, Namibia, which is over 80 times larger than what Schubert et al. (2009) found (Table 4). As additional home range data for other species were published (Table 4), sengi home ranges of a square kilometer seemed almost unbelievable.

Table 4 Comparison of home range areas for different sengi species as determined by different methods and reported in the literature.

See Fig. 6 for full species names. Mean weight (g) and mean rainfall (mm) column based on data from references, or other literature. The tilde (∼) indicates values are not calculated means, but an estimate for various reasons (see text). Mean areas (ha) are presented for sexes combined (C), but if the datum was not provided, then we calculated the mean of the two sexes. Male only (M), and females only (F). Number of individuals used to calculate mean areas for the sexes are in parentheses (M/F). Home range areas in BOLD font are used in comparing mean home range areas for sengis with study site mean yearly rainfall and mean body weight (Fig. 6). See methods section for explanation of inter-home-range distances.

Species	Weight rainfall	100% convex	95% convex	OREP	95% kernel	Inter-home- range distances	Reference	
M. micus	26.9 g ∼10 mm	14.92 C (5)	11.64 C (5)	7.21 C (5)	22.76	425 m	This study	
M. flav	31.5 g 24 mm	∼9.0 (?)	–	–	–	300 m	Sauer, 1973	
M. prob	∼50 g 160 mm	1.25 C 1.7 M 0.8 F (23/24)	–	–	–	–	Schubert et al., 2009	
E. intufi	46.0 g 293 mm	–	0.47 C 0.61 M 0.34 F (7/7)	–	–	–	Rathbun & Rathbun, 2006	
E. brachy	∼45 g 650 mm	–	–	–	0.33 C 0.41 M 0.25 F (4/5)	–	Yarnell et al., 2008	
E. myur	60.0 g ∼730 mm	0.30 C 0.39 M 0.20 F (6/6)	–	–	–	–	Ribble & Perrin, 2005	
E. myur	∼60 g 315 mm	1.06 C (4)	–	–	–	–	Olbricht, Sliwa & Abenant, 2012	
E. ruf	58 g 640 mm	0.34 C (10)	–	–	–	–	Rathbun, 1979	
P. tetra	∼200 g ∼800 mm	1.2 C (14)	–	–	–	–	FitzGibbon, 1995	
P. tetra	196 g ∼700 mm	–	0.95 C 1.2 M 0.7 F (4/6)	–	–	–	Oxenham & Perrin, 2009	
R. chrsyo	∼500 g ∼1,000 mm	4.1 C (28)	–	–	–	–	FitzGibbon, 1995	
R. chryso	540 g 1,040 mm	1.7 C (11)	–	–	–	–	Rathbun, 1979	

To better understand the Sauer (1973) home range estimate, we closely examined his definitions of space use, which are different than what is typically used. For example, Sauer (1973, pages 74 and 94 among others) states that M. flavicaudatus had an average home range of a square kilometer, but he also indicates that this was in fact a crude calculation of density based on his 20 sq km study area. On the other hand, based on his descriptions, it is almost certain that his sengis had home ranges that were larger than other small sengis, which are less than 2 ha (Table 4). To obtain a crude home range estimate that more closely conforms to the more widely accepted definition of Burt (1943), we used the 300 m mode of the average distances between the main shelters used by two closest neighbors on adjacent home ranges (Sauer, 1973, pg 71, Table 1, Fig. 7, pg 95). If we assume that the length of each of two adjacent sides of a hypothetical home range is thus 300 m, and adjoining home ranges were in relatively homogeneous habitats (Ibid.), we obtain an estimated home range area of about 9 ha, which is over an order of magnitude smaller than the density of 100 ha/sengi that he called a “home range”. This re-estimation of home range size (sensu Burt, 1943) reconciles the seemingly small modal average distances between shelters on adjacent home ranges of 300 m and an estimate of home range areas of 1 sq km. Lastly, 300 m is not very different from our similar metric of 425 m for M. micus (Table 4). Even through our home range estimate for M. flavicaudatus is not strictly comparable with other sengi home range estimates, it probably is sufficient for our discussion.

The literature related to mammalian home ranges is large, including attempts to relate the sizes of home ranges with physiological factors such as trophic level (calorie sources), body size (calories needs), metabolic rate (rate that calories are used), social structure (group versus individual needs), and phylogeny (McNab, 2002, pages 335–336). These factors are highly variable across a wide range of mammals, making comparisons difficult, except for the sengis, which share a very tightly defined adaptive syndrome with very similar phylogeny, metabolic rate, morphology, diet, reproduction, locomotion, social structure, etc. (Rathbun, 1979; Rathbun, 2009). In contrast, the variation in the body size and habitats occupied by sengis stands out (Rathbun, 2009, see ‘Introduction’).

Although body weight data for sengis are available (Table 4), there are several metrics that might be used to quantify the habitats used by sengis. Given the life history traits of sengis, we believe that prey abundance is particularly important. Unfortunately, prey abundance is not easily measured or available from most sengi study sites (but see Rathbun, 1979; FitzGibbon, 1995). However, rainfall is probably a reasonable proxy, and these data are available (Table 4). When sengi home range sizes are plotted against rainfall, the points for M. flavicaudatus and M. micus are far removed from the rest of the sengis in the plot (Fig. 6A), despite our using conservative estimates for these two species (see discussion of Sauer above, and Table 2). Although M. proboscideus occupies a low-rainfall habitat similar to its congeners, it clusters with the other smaller sengis (Fig. 6A), which suggests that low rainfall habitats do not fully explain home range size for similarly sized sengis (keeping in mind their very similar adaptive syndrome). The Succulent Karoo, where the data for M. proboscideus were gathered (Schubert et al., 2009; Schubert, 2011), is a relatively small area between the very low and concentrated winter rainfall regime of the Namib Desert to the north, and the low summer rainfall regime of the Mediterranean climate to the south. Although the Succulent Karoo is arid, the rainfall is spread across both winter and summer months (Desmet & Cowling, 1999). This rainfall pattern results in a richer vegetation (Cowling & Hilton-Taylor, 1999) and invertebrate fauna (Vernon, 1999) than might be expected based only on total average rainfall. Thus, the home range area of M. proboscideus clusters closer to the other small sengis than with M. flavicaudatus and M. micus (Fig. 6A). The general positive relationship of mammalian body weight and home range size (McNab, 2002) is supported by syntopic Petrodromus and Rhynchocyon in a coastal forest in Kenya (FitzGibbon, 1995; Table 4; Fig. 6B), but is overshadowed by the Macroscelides species, especially the two in the Namib Desert (Fig. 6B). Based on our analysis, we hypothesize that prey availability may have the greatest influence on the home range sizes of sengis, but unfortunately data on prey availability are lacking for most sengis.

Figure 6 Scatter plots of sengi home range areas against study site rainfall (A) and sengi weights (B).

Data from this study and published literature (Table 4).

In most sengi home range studies, a male will occasionally attempt to overlap with more than one female, often resulting in an exceptionally large and oblong home range. However, this configuration is not stable due to the would-be polygamous male retreating when a new male appears to associate with (and mate-guard) one of the females (Komer & Brotherton, 1997; references in Table 4). We speculate that the large hour-glass-shaped home range of #4254M represented a similar attempt at polygamy (although we did not trap the northwestern end of his home range to determine if there was a female in that area).

Male–female sengi pairs exhibit few pair-bond behaviors and spend relatively little time together (except during brief periods of estrus), yet some species have home ranges that are virtually congruent (Rathbun, 1979), while in others the ranges only partially overlap (all other references in Table 4 and Fig. 3). One hypothesis to explain this variation is that the degree of overlap is density dependent. In habitats where sengis essentially occupy all suitable space and thus are dense, male and female home ranges are nearly congruent and intra-sex overlaps are rare because the areas are defended sex-specifically, whereas when sengis are more dispersed, the home range overlap within male–female pairs is reduced (Rathbun & Rathbun, 2006). At our Namibia study site, we were unable to capture and radio-track as many sengis as we had hoped, and we could not document that all resident sengis were radio-tagged. However, based on our intensive trapping effort and extensive radio-tracking and observation activities on the study site, it is unlikely that many if any resident sengis escaped our notice. We speculate that the home range configurations that we documented (Fig. 3) are consistent with the density dependent home range overlap model. We again hypothesize that prey availability may be the most important underlying factor in determining sengi density, and thus many home range characteristics. However, multiple factors may be involved (Di Stefano et al., 2011), including mate availability. Sauer (1973) suggested that shelters may have been limiting for M. flavicaudatus, although we doubt this was the case at our study site (Fig. 1).

Sheltering

There were two noteworthy findings regarding shelters. First, was the lack of a central or home burrow or shelter, as found in many other small mammals. Second, was how unremarkable the shelters were; there was no sign of bedding, excavation or alteration, and every shelter seemed to simply be a small space or crevice under a rock where sengis hid during the day. Both findings are similar to the sheltering habits of other Macroscelidinae (Rathbun, 2009). It is difficult to determine which factors were motivating the use of rock shelters by M. micus because of the large number of possible factors, including sengi behavior, predation threat, weather, environmental conditions, and shelter availability. We believe the most important two factors were the thermal traits of the shelters and predation threat.

We found that midday temperatures of shelter rocks were inversely related to rock thickness (confirming the ability of thicker rocks to resist temperature fluctuations), and we found that shelter openings were significantly orientated toward 193° south. We suspect that these two features have related consequences for sheltering sengis. Like many deserts, the Namib is characterized by frequent high and low temperature extremes (Seely, 2004). Thus, the size and orientation of a shelter rock may allow M. micus to passively (behaviorally) avoid temperature extremes and thus reduce energy needed for thermoregulation (McNab, 2002), which is likely important for such a small-bodied desert dweller. For example, the sengis might choose shelters in order to take advantage of the thermal inertia of rock to buffer day and night temperature extremes. In western Namibia, the prevailing winds come from the south (Mendelsohn et al., 2002). Winds often blew hard (we measured up to 13.5 m/s) during the midday and afternoon. Thus, south-facing shelter entrances may be more exposed to cooling breezes during the heat of the day. In addition, the south side of a shelter corresponds with the shady and thus cooler side of the rock during the heat of midday because the sun is slightly angled toward the north during this time of year.

Lovegrove, Lawe & Roxburgh (1999) documented daily torpor in M. proboscideus, which is likely a physiological strategy that sengis use under conditions of limited food availability and low temperatures to conserve energy (Mzilikazi, Lovegrove & Ribble, 2002). It is possible that the torpid sengi we encountered (#4020F) was implementing this strategy in this hyper-arid study site with a hypothesized low abundance of prey. However, more research is needed to further explore the relationships between shelter traits, shelter choice, and sengi behavior.

Sauer (1973) also believed that thermoregulation was an important feature of the shelters that were used by M. flavicaudatus. However, potential shelters were less abundant at Sauer’s study site compared to our site, as clearly illustrated by his numerous figures (Ibid.). Low shelter availability may also partially explain why M. flavicaudatus either uses abandoned rodent burrows, or excavates shallow shelters in the gravel substrates (Sauer, 1973). The only hint that M. micus might excavate shallow shelters was the use of two shallow holes (9 and 22 cm deep) by the two young sengis that we captured. We had no direct evidence that sengis fashioned these sites, so they may have been abandoned rodent burrows, although rodents were even less common than sengis at our study site.

Predation is often difficult to document, but one of our collared sengis was preyed upon, possibly by a Cape Fox, suggesting that avoiding predation may be challenging. Thus, one important feature of shelters is likely the availability of refuges that provide adequate protection from predators. Perhaps just as important is the use of multiple shelters with a very low rate of return to any single shelter, and the lack of feces accumulation in the shelters. These behaviors may be related to reducing visual or olfactory cues that predators use to find sengis. Related explanations were proposed for the similar spatial and temporal traits of sheltering sites of Elephantulus intufi (Rathbun & Rathbun, 2006), and also the nesting traits of Rhynchocyon (Rathbun, 1979). Additionally, at our study site there was little cover other than under rocks. This may explain the strictly nocturnal behavior of M. micus, which effectively would avoid predation by the numerous diurnal predators, including several raptors and bustards.

Sengi adaptive syndrome

We found that M. micus largely conformed to the life history features characteristic of other sengi species, especially the Macroscelidinae, including swift and agile cursorial locomotion, relatively exposed multiple sheltering sites, and possibly spatial organization. Additionally, M. micus has small litters of precocial young (Dumbacher et al., 2014). However, we were unable to confirm whether M. micus has a female absentee maternal care system, and whether its diet was composed mainly of small invertebrates, as it almost surely is based on the near absence of any other visible food at our study site and its morphology, which is very similar to other sengis that are known to prey on invertebrates (Rathbun, 2009).

There are several behavioral features that are worth discussing for future comparative studies, but may only be peripheral to the adaptive syndrome. We failed to find any indication that M. micus created trails on the substrate, as do other sengis (Rathbun, 1979), including M. flavicaudatus (Sauer, 1973). We suspect that M. micus used familiar travel routes, but paths did not form because the substrate was dominated by rock. If sengis traveled mainly on familiar routes, we probably only trapped them when a route coincided with a trap location, which might explain our low capture rate. We also failed to see or hear foot-drumming during stressful situations, including while in live traps, which is also characteristic of other sengis (Rathbun, 1979; Faurie, Dempster & Perrin, 1996). Neither distinctive latrines of dung pellets (Rathbun, 1979), nor scent-marking behaviors (Rathbun, 1979; Faurie & Perrin, 1995) were observed during our study, despite M. micus having a very large subcaudal scent gland (Dumbacher et al., 2014). Daily torpor, which is an energy conservation strategy in M. proboscideus (Lovegrove, Lawe & Roxburgh, 1999), may be used by M. micus, based on our observation of a torpid individual.

Conclusions

The home range pattern that emerged from our study was similar to the findings for other sengis, except that the areas of M. micus were exceptionally large. Their size was likely the result of low rainfall, sparse vegetation, and low densities of invertebrate prey. The home range characteristics that we found are similar to those of socially monogamous sengis, suggesting that M. micus may also be socially monogamous, although the highly dispersed individuals made this difficult to establish. In nearly all aspects, M. micus conformed to the sengi adaptive syndrome, although with some variation to accommodate desert conditions, such as sheltering habits to buffer desert temperatures. Their sheltering patterns also may have evolved to prevent the accumulation of olfactory and visual cues used by predators. Their nocturnal activity may also be related to predator avoidance in a desert with little plant cover.

Supplemental Information

Data S1 Location Data derived from radio-tracking

Data associated with locations for all radio-tracked sengis.

Click here for additional data file.

Supplemental Information 1 Home range, rainfall, weight data

Data used to construct scatter diagrams for sengi home ranges versus weights and also home ranges versus rainfall.

Click here for additional data file.

Data S2 Shelter data for radio-tracked sengis

Physical features associated with shelters used by radio tracked sengis.

Click here for additional data file.

Tim Osborne of Windpoort Farm and Seth Eiseb of the University of Namibia provided greatly appreciated logistical and field support, as did Roger Fussel and Lindy van den Bosch of Big Sky Lodges in Windhoek. Eugene Marais at the National Museum of Namibia loaned us some live traps. We appreciate Hubert Hester of Kalahari Car Hire meeting our vehicle needs. The careful translation of Sauer (1973) into English by Nani Croze in 1975 in Nairobi, Kenya, continues to be invaluable. Our wives Carolyn Rathbun and Tiffany Bozic (and daughter Tesia) provided invaluable logistical and moral support while we were all in Namibia. We thank David Ribble for his insightful comments on an early version of this paper, as well as all the useful comments and suggestions provided by an anonymous peer reviewer and Julian Di Stefano, and Academic Editor Donald Kramer.

Additional Information and Declarations

Competing Interests

Author Contributions

Animal Ethics

Field Study Permissions

The authors declare there are no competing interests.

Galen B. Rathbun and John P. Dumbacher conceived and designed the experiments, performed the experiments, analyzed the data, contributed reagents/materials/analysis tools, wrote the paper, prepared figures and/or tables, reviewed drafts of the paper.

The following information was supplied relating to ethical approvals (i.e., approving body and any reference numbers):

California Academy of Sciences Institutional Animal Care and Use Committee (approval number 2014-1).

The following information was supplied relating to field study approvals (i.e., approving body and any reference numbers):

Namibia Ministry of Environment and Tourism (permit number 1927/2014).

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
