# Peer review of "Home range and use of diurnal shelters by the Etendeka round-eared sengi, a newly discovered Namibian endemic desert mammal"

_PeerJ, doi:10.7717/peerj.1302_

## Round 0.1 · original submission · Minor Revisions

Both reviewers found this a fundamentally sound, though limited, study of home range and shelter characteristics of a previously undescribed species of sengi. Their comments relate to the organization, style and detail to be included in your presentation. I agree with these views. Therefore, the changes required before publication are considered minor revisions. Both of the reviewers have provided valuable comments. One of my responsibilities as editor is to provide some guidance with regard to their suggestions. Below I comment on a few of their suggestions where opinions may vary.

The reviewers differ in their response to your prose, one considering it generally lively and engaging, the other excessively detailed and wordy. While I agree with reviewer 2 that the manuscript currently contains detail and anecdotal material that might not be permitted in a traditional article, I personally found it interesting rather than tedious. I thought that even with a limited sample size, your comparative examination of home range areas made a useful contribution. Given that PeerJ does not publish a paper version, excess length is principally a problem of reader engagement. I therefore suggest that you carefully read all the suggestions and make your own decisions as to what should be excluded.

I do agree that the detailed description of the Sauer's work seemed wordy, out of place, and perhaps condescending in parts. You might consider trying to make the same points in less personal way.

I also agree that you have very little evidence of having tracked all individuals in your area. Thus, the home range overlap and distance between home range centers add very little to the overall contribution and could be eliminated or just briefly mentioned.

Given the scope of your study, I do not think you need more literature on the relationship of food availability to home range size. However, you might consider referencing the specific pages in McNab's book where this relationship is reviewed.

I do not feel that you need confidence intervals on your regression lines, given that all data points are visible and you have provided the r-square.

Finally, I don't agree that your revision of the Introduction should present testable predictions if this was not the way you initiated your study. Adding post hoc predictions provides a misleading representation of the scientific process and making comparative predictions incorrectly implies that your sample size would be adequate to statistically support inter-species differences.

Additional comments and suggestions from the editor:

L94. Metric units preferred
L241. I was pleased to see the inference that a sengi unfamiliar with its home range made errors while fleeing and was easy to catch because some time ago I experimentally documented somewhat similar effects in eastern chipmunks (M.F. Clarke et al 1993 Oikos 66:533-). Nevertheless, such patterns might also occur if the animal was injured or unwell, particularly if it was experiencing a visual deficit. I am not aware of much subsequent documentation of effects of familiarity on home-range navigation. Thus, it is not fully 'obvious' that a stumbling animal is unfamiliar with the area in which it is traveling.
L319. Use consistent punctuation for time.
L448. I am not sure whether you are using the term 'search image' correctly. Search image is an inferred neurobiological constraint that prevents animals from searching for more than one type of cryptic prey at a time (e.g., Dukas & Kamil 2001, Behav Ecol 12:192-). If you mean that foxes would learn about the types of shelters used by sengis from the odor of previously used shelters and use this information to search for sengis in shelters in the future, I think that the usage would be correct. If you mean only that limited use of shelters minimizes olfactory cues that could be used by foxes to detect them, search images would not be involved. The latter seems a more straightforward concept, given the lack of evidence of any aspect of the predator's behavior.
L502. Do you have a reference for the inference of an insect diet based on morphology? If not, you need to provide at least some supporting evidence. Also, I don't think the commas in this sentence are correctly placed.
Table 1. The basis for the order of individuals in this table is not clear. Wouldn't it make more sense to order them by tagging date?

·

Basic reporting

Basic reporting is sound, except I thought the introduction could be restructured and expanded to better set the context, introduce related home range size literature and theory, and develop some testable predictions.

Experimental design

The experimental design and methods are generally sound. I think the research question can be better linked to theory and existing research (see Basic reporting).

Validity of the findings

The findings are valid and data have been made available. A small sample size limits generalizations that can be drawn from the findings. I think the findings can be better linked to theory, past research and predictions, all of which need to be explained and developed in the introduction.

Additional comments

This is a very well written manuscript that provides some initial data about movement ecology and shelter use of the Etendeka round-eared sengi. You have a lovely turn of phrase – for example, phrases like 'The sengi easily kept ahead of us as it bounded from rock to rock' and 'The sengi was obviously unfamiliar with the area because he continually stumbled over and bumped into rocks as he clumsily fled' paint a vivid picture of events and makes the manuscript more engaging and interesting to read.

I do, however, have several suggestions that I think will improve the manuscript. These are predominantly related to the content and structure of the introduction and discussion. Major points are outlined below, and I’ve made a range of other comments using track changes directly in the manuscript.

Introduction

As it stands, the introduction is very descriptive and doesn’t really pose a testable hypothesis or prediction. Given the objective stated on lines 67-68 I suggest structuring the introduction as follows:

1. Begin more broadly, perhaps introducing the concept of adaptive syndrome and explaining what it means ecologically.

2. Describe the sengis, as you have done

3. Make some predictions about the focal life history traits of the new species based on the adaptive syndrome and phylogenetic constraints seen in other species. I guess the predictions should relate to movement (home range) and shelter as these are the two main thinks you have data about.

I think this would make for a stronger introduction, ending in a set of testable predictions.

Discussion

I found the discussion oddly organised with some unexpected content.

First, I suggest beginning the discussion more broadly, perhaps by reiterating the general purpose and value of the research.

Second, I found the two paragraphs near the start (lines 329-364) focusing on the inadequacies of a study by Sauer (1973) distracting and a bit irrelevant. The sole purpose of this longish diatribe seems to be to justify the data point in Fig 6. marking the home range size of M. flavicaudatus. Would it not be sufficient to say, in a few sentences, that due to suspect methods the home range size of M. flavicaudatus is unknown, but likely to be much larger than that of other species (with the exception of M. micus).

Third, a fair bit of the discussion about home range builds an argument that the dominant influence on home range size is food availability. While this makes sense, reference to the large literature on this topic is largely missing (you appear to use McNab (2002) as a surrogate). Also, if this is an argument you want to make, I suggest you build this into the introduction. Because of the low rainfall at your study site, and the assumed low abundance of prey, you could reasonably predict that M. micus would have a large home range. Harestad and Bunnell (1979) might be useful here – Ecology 60: 389-402.

Also, limitations of your very small sample size should be noted somewhere in the discussion. This really is a very preliminary study with data on only five individuals.

Reviewer 2 ·

Basic reporting

Data are available as draft supplements. Animal care/use approval is listed appropriately. The submission is self-contained and an appropriate unit of publication.

The background for the study is thoroughly presented for the most part. One very easily correctable issue is the following: Without reviewing the references, it should be made more obvious to the reader in the intro whether there are any past studies on home range/movement on sengi populations that may have included this newly described species. I.e. "newly described" does not automatically mean completely unstudied or newly discovered. Clarifying this in a sentence or phrase would help the reader better understand the value of the information produced by the study.

The manuscript is well written but still has a "not final draft" feel to it. The manuscript is overly wordy in some places, a distracting problem that should be corrected through thorough editing. Please see General Comments.

Experimental design

In contrast to many overly wordy areas, the last paragraph of the intro is overly brief. Expand and cover the key questions more specifically. Quantifying these parameters, and comparing through lit review... Any a priori expectations?

I missed anything in the methods about how often locations were obtained per individual and how radio-telemetry was planned to get a representative, systematic set of locations for each animal each night, which usually is tied to a systematic design for rotating through individuals and times. This is a core detail of the methods that determines the value of the data.

The value of the overlap analysis for a few individuals depends on the assumption that one has captured nearly all local individuals (i.e. who cares if two individuals don't overlap much with each other if they each overlap much more with uncaptured individuals). I'm not convinced that this is the case, and I don't see the results or discussion regarding overlap analyses or measures of dispersion as being conclusive enough to be publishable. I think the basic summaries of movements, home range size estimates, and shelter use have value, but going beyond these basic summaries is not justifiable. The sample size is far too small to offer anything more than an initial snapshot of space/habitat use of the species. Without more information, comparisons with other species in the group and evaluation of conformity with the adaptive syndrome with the group are highly speculative. Information from a handful of animals for a few weeks is noteworthy for an unstudied species - but that doesn't make it a foundation for broad-scale inference about whether that species meets the expected patterns for its taxonomic group.

On a much lesser note: I dislike the unsubstantiated statements about the OREP/concave polygon estimators (e.g. lines 153-156, lines 339-347). The history of home estimators/models is full of such "well, this looks good for my data sets so it should be used"-type statements without extensive evaluation of the expected accuracy of the technique under a broad range of realistic conditions." Fairly small data sets for a few individuals of one species for a few weeks is not a basis for saying much about an estimator, and these statements should be deleted.

Validity of the findings

The sample size of animals and short duration of the study (a few weeks) are major limitations of the study. This sample size comes close to being unpublishable in comparison with current standards for space-use studies in these different aspects of sample size. Essentially this is a pilot study for developing more conclusive movement/space use studies of the species. Given the novelty of the data on a newly described species, I think the the core information makes for a meaningful natural history contribution, but I think the manuscript is on very thin ice when it goes beyond basic data summaries of home-range estimates and shelter characteristics. Discussion of overlap and comparisons with other species in this group are not anywhere close to being conclusive.

Beyond this, the word:information ratio is far too high, given the handful of animals studied for a small period of time. For example, lines 220-255 make a full page for a few incidental observations from the study, with lots of random story telling that does not strike me as having general value for understanding the species. Despite the authors' expertise for this group, it was difficult to read the discussion: the data at hand warrant 3-4 paragraphs of discussion, not several pages.

Additional comments

I recommend that the core data (and basic estimates of home range size) have value as a contribution, but that to be publishable the manuscript should focus on reporting these core data without attempting to draw general conclusions about the space use of the species on its own or in comparison to other sengis, except as a brief step in setting up more conclusive studies. The core results have validity, but too much of the paper is exploratory and speculative given the small sample size.

To be brought within the range of standard scientific writing, the manuscript needs straightforward but significant editing for brevity and conciseness, and flow.

Intro: Paragraph 2 of the intro needs to be broken into two paragraphs - too much information crammed into one long paragraph.

Please consider phrases and sentences such as: Line 82: "made walking difficult", lines 82-84: towns and driving distance to towns; lines 86-87: details of fog drying:; lines 121-123: a full sentence about the format for animal ID numbers. These are examples of details that are superfluous, to a degree not consistent with what one expects in published scientific studies. There are too many details not critical for understanding the results / limitations or for replicating the study. (The clear study-site pictures in the figures give an adequate picture of the terrain and remoteness without needing any added text in the manuscript body.)

Lines 112-115: Wordy and unclear.. How about "We used bright... to spot eartagged sengis, frequently with the aid of binoculars. Tags were visible at least 100 m line of sight but fog, dust, and rocks often reduced visibility to much lower distances."

Lines 127-131: Edit for brevity. "We homed in until we saw the active animal and recorded the location where it was first sighted. If it was sheltering under a rock, we located the shelter and recorded the location."

LInes 137-141: These are pretty irrelevant details that have no bearing as to the quality or reproducibility of the study. The care in testing the receivers is nice but this is far too much text for an accuracy trial consisting of 26 test locations! This paragraph should be boiled down to one or two sentences.

Lines 171-172: Unnecessary sentence

LInes 180-194: All well written and appropriate.

---

## Round 0.2 · accepted · Accept

The revisions are appropriate, and the manuscript is now suitable for publication.